# ATLAS-ALIGNMENT: MAKING INTERPRETABILITY TRANSFERABLE ACROSS LANGUAGE MODELS

## ABSTRACT

Interpretability is crucial for building safe, reliable, and controllable language models, yet existing interpretability pipelines remain costly and difficult to scale. Interpreting a new model typically requires costly training of model-specific sparse autoencoders, manual or semi-automated labeling of SAE components, and their subsequent validation. We introduce **Atlas-Alignment**, a framework for transferring interpretability across language models by aligning unknown latent spaces to a Concept Atlas — a labeled, human-interpretable latent space — using only shared inputs and lightweight representational alignment techniques. Once aligned, this enables two key capabilities in previously opaque models: (1) semantic feature search and retrieval, and (2) steering generation along human-interpretable atlas concepts. Through quantitative and qualitative evaluations, we show that simple representational alignment methods enable robust semantic retrieval and steerable generation without the need for labeled concept data. **Atlas-Alignment** thus amortizes the cost of explainable AI and mechanistic interpretability: by investing in one high-quality Concept Atlas, we can make many new models transparent and controllable at minimal marginal cost.

## 1 INTRODUCTION

Large language models (LLMs) are increasingly deployed in domains where safety, reliability, and controllability are critical. Yet, their internal representations and processes remain largely opaque to their users, hindering verifiability and trust. Model activations capture the semantic and functional structure of processing, but without the tools to interpret them, we cannot rigorously assess how a model arrives at its outputs or intervene when it is behaving in unforeseen ways. Interpretability is thus essential for both trust and reliability, as well as for practical control of model behavior.

Advances in mechanistic interpretability and explainable AI have begun to uncover these latent structures using sparse autoencoders (SAEs) (Bricken et al., 2023) and automated feature-discovery pipelines (Bills et al., 2023; Choi et al., 2024; Dreyer et al., 2025). These can, for instance, extract latent features that are both monosemantic and can be described in natural language through human labeling. Such features enable analysis of reasoning processes and make controlled interventions on model activations easier. However, current interpretability methods remain costly and difficult to scale. Each new model and layer requires training SAEs, generating feature descriptions, and validating them individually. The need to explain each new model variant from scratch makes comprehensive interpretability computationally expensive and often infeasible.

In this work, we pursue a complementary direction: rather than interpreting each model in isolation, we ask whether interpretability can be transferred across models. We introduce **Atlas-Alignment**, a framework for aligning the latent space of an unknown "subject model" to a well-understood, human-labeled latent space that we refer to as Concept Atlas. Once aligned, the subject model inherits the interpretability of the Concept Atlas: its features can be semantically queried, compared, and steered without the need for costly SAE training or labeled concept datasets.

**Atlas-Alignment** builds on two complementary hypotheses. The Linear Representation Hypothesis suggests that semantic concepts are often linearly encoded as directions in latent spaces (Park et al., 2024), while the Platonic Representation Hypothesis suggests that different LLMs converge on broadly similar latent structures (Huh et al., 2024). Together, these imply that a single, carefully constructed Concept Atlas could serve as a universal "concept hub". By aligning subject models

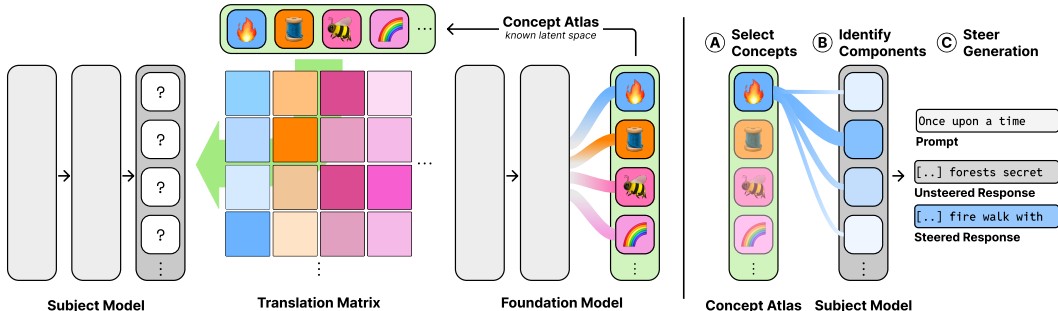

Figure 1: **Atlas-Alignment** makes the latent space of a subject model interpretable by aligning it with a Concept Atlas — a human-interpretable, labeled latent space. **Left**: The subject model's hidden representations are mapped into the Concept Atlas, allowing each subject feature to be described as a linear combination of atlas concepts. **Right**: Once aligned, the method enables a range of interpretability tasks. (A) One or multiple concepts are selected from the Atlas, (B) corresponding subject model components are identified, or (C) the subject model's output is steered along the concept direction.

to this atlas using only shared input data and lightweight transformations, we can recover semantic structure and enable plug-and-play interpretability across a wide range of models.

This translation unlocks several capabilities. Aligned models support semantic search and grouping of features, cross-model and cross-layer comparison of representations, and controllable steering of generation along human-interpretable directions, allowing us to navigate yet unexplored latent spaces — all without training probes or SAEs, or generating synthetic datasets. Crucially, the cost of interpretability is amortized: a single high-quality Concept Atlas can make many new models transparent and steerable at a minimal marginal cost. Although in this work we mainly interact with features of the MLP- and residual stream layers, due to the existence of high-quality labeled latent spaces in these domains, we believe a promising application of our framework lies in the translation of attention heads, which often perform specialized tasks such as context retrieval within Transformer models (Kahardipraja et al., 2025).

Our contributions are as follows:

1. We introduce **Atlas-Alignment**, a simple and general framework for translating between feature spaces in language models using Concept Atlases, shared input data, and lightweight representational alignment methods.

2. We demonstrate the practical applications of **Atlas-Alignment** for semantic feature search and semantic steering without the need for labeled concept-data.

3. We provide a quantitative evaluation of how various representational alignment methods perform in our framework, measuring their translation quality, semantic retrieval performance, and controllability in steering.

In the following sections, we first review related work on interpretability and representational alignment (Section 2). We then introduce the **Atlas-Alignment** framework, including the construction of Concept Atlases and the methods used for latent space translation (Section 3). Next, we present experimental results demonstrating semantic feature identification, cross-model retrieval, and concept steering (Section 4). Finally, we discuss the implications and limitations of our approach (Section 5).

## 2 RELATED WORK

**Features and Concepts** Neural network neurons and their linear combinations have been shown to encode human-aligned concepts to a surprising degree (Achtibat et al., 2023; Bykov et al., 2023). Supervised methods for identifying such features typically rely on curated concept datasets to locate directions or neurons of interest (Kim et al., 2018; Bau et al., 2017). Although effective, these meth-

ods are inherently limited to the pre-defined choice of concepts. Unsupervised approaches instead examine activations from large, unlabeled datasets to discover semantic clusters. A particularly influential line of work has been the development of sparse autoencoders (SAEs) (Bricken et al., 2023), which decompose polysemantic features into sparse, more monosemantic ones, which are often highly interpretable.

This progress has increased the need for methods that assign meaning to features in a scalable manner. Recent work automates the generation of natural language descriptions of features (Bills et al., 2023; Paulo et al., 2024; Templeton et al., 2024), along with evaluation methods to assess their quality (Bills et al., 2023; Kopf et al., 2024; Puri et al., 2025; Gur-Arieh et al., 2025). However, these pipelines remain costly: each new latent space typically requires training SAEs, producing feature descriptions, and validating them before meaningful semantic interaction becomes possible.

A complementary direction leverages existing latent spaces to interpret others. For instance, CLIP embeddings (Radford et al., 2021) have been used to describe or query vision model features in semantic terms (Oikarinen & Weng, 2023; Dreyer et al., 2025).

Our approach, **Atlas-Alignment**, extends this idea. We exploit an interpretable SAE latent space and transfer its semantics into a subject model we wish to understand. This enables semantic interpretation and steering of features without fine-grained concept datasets or costly re-interpretation of each new latent space.

**Aligned Latent Representations** Since our goal is to translate concepts across latent spaces, a natural question is whether two models' latent spaces actually represent concepts in a transferable way. Here, the Platonic Representation Hypothesis (Huh et al., 2024) provides a theoretical motivation: it posits that, as models scale in size, data, and task diversity, they converge on a shared model of reality, resulting in similar latent spaces across architectures and even modalities. Complementarily, the linear representation hypothesis (Park et al., 2024) argues that many human-aligned concepts are encoded approximately linearly in activation space. Although these hypotheses will not hold perfectly in practice, together they provide a theoretical basis for meaningfully transferring concepts across the latent spaces of different models. Additionally, the evaluation procedures we introduce make it easy to assess how well this alignment holds for a given model pair, rather than assuming the hypotheses apply universally.

Building on these hypotheses, several works have approached cross-model alignment from a practical perspective. Jha et al. (2025) train mappings between models and a shared latent space using an unsupervised cycle consistency loss approach, while Thasarathan et al. (2025) propose training SAEs that jointly decompose representations from multiple models, creating a shared backbone concept space. However, both approaches require expensive training and can additionally suffer from high reconstruction loss.

Closely related to our work, Maiorca et al. (2023) introduce a component-stitching approach that uses lightweight linear transformations to reuse components from other models to improve performance on downstream tasks such as classification or reconstruction. Huang et al. (2025) study cross-model steering by constructing supervised steering vectors and transferring them across models via a linear mapping learned from small contrastive datasets. While these approaches align representations to support task reuse or supervised steering, our method instead maps a model's internal representations into a Concept Atlas, a sparse, labeled reference space built for interpretation. This enables the transfer of mechanistic interpretability, allowing semantic search and dataset-free controllable steering in new, opaque models without retraining or relabeling interpretability tools like SAEs.

## 3 METHOD

Our goal is to align the latent representations of a subject model (of which we have no prior understanding) with a Concept Atlas, a labeled and interpretable latent space derived from a foundation model. Once aligned, the subject model inherits the interpretability of the atlas: its features can be queried, compared, and modified along semantically meaningful directions.

This alignment requires only shared input data: by presenting the same dataset to both models and comparing activations, we can construct lightweight mappings that reveal the semantic structure of otherwise uninterpretable features.

### 3.1 BACKGROUND AND NOTATION

Let $\mathbf{X} = \{x_1, \ldots, x_N\}$ be a dataset, where each sample $x_i$ is a sequence of text. The subject model $f_s : \mathcal{X} \to \mathcal{H}_s$, maps from data domain $\mathcal{X}$ into an intermediate feature space $\mathcal{H}_s$, an unknown space we aim to interpret. The foundation model $f_f : \mathcal{X} \to \mathcal{C}$ maps from the data domain into our Concept Atlas $\mathcal{C}$, a feature space that is semantically interpretable. Forwarding the dataset through the models and applying max-pooling over the sequence lengths, we retrieve aggregated activation matrices $A_s \in \mathbb{R}^{N \times d_s}$ and $A_c \in \mathbb{R}^{N \times d_c}$. We use pooling to handle differences in how the two models tokenize the input, and choose max-pooling as it highlights the strongest per-sequence activations. A comparison with mean-pooling is given in Section 4.3.1.

### 3.2 CONCEPT ATLAS

The Concept Atlas is our reference space of interpretable features, where each dimension corresponds to a human-understandable concept. We rely on SAEs to construct our Concept Atlas due to their strength in producing sparse, monosemantic and human-interpretable latent spaces. Each Concept Atlas feature $c_k \in \mathcal{C}$ can be assigned a natural language description $d_k \in \mathcal{D}$ via manual or automatic labeling methods. This annotated atlas serves as a "semantic dictionary", where every feature corresponds to a concept humans can name and reason about. We can combine multiple features and weight them according to the concepts we want to identify or steer. Aligning new models to this space lets us carry over those semantics without repeating the costly process of building and labeling SAEs from scratch.

### 3.3 TRANSLATING LATENT SPACES

We define a translation function that expresses subject model features of a layer in terms of the Concept Atlas features

$$\text{translate} : \mathbb{R}^{N \times d_s} \times \mathbb{R}^{N \times d_c} \to \mathbb{R}^{d_s \times d_c}, \ (A_s, A_c) \mapsto \mathbf{T}_{s \to c} \tag{1}$$

Each row of the resulting matrix $\mathbf{T}_{s \to c}$ represents a subject model feature in terms of the Concept Atlas features. This function can be instantiated with standard representational alignment methods. As an example, we show the Orthogonal Procrustes method. All methods used in this work are described in the Appendix A.3.

**Orthogonal Procrustes Translation:** Constrains the translation matrix to be orthogonal, so the alignment is a pure rotation or reflection of the space

$$\text{translate}_{\text{OP}}(A_s, A_c) = \underset{\mathbf{T}_{s \to c}}{\arg\min} \|A_s - A_c \mathbf{T}_{s \to c}^\top\|_F^2 \quad \text{s.t.} \quad \mathbf{T}_{s \to c} \mathbf{T}_{s \to c}^\top = \mathbf{I} \tag{2}$$

After a row-wise $L_2$-normalization of the activations, this is equivalent to minimizing the cosine distance between activations.

### 3.4 USING LATENT SPACE TRANSLATIONS

Once the mapping is learned, we can use the matrix $\mathbf{T}_{s \to c}$ and the knowledge encoded in the Concept Atlas to make the subject model latent space both interpretable and controllable. This involves three steps: Creating a Concept Query, mapping it into the subject model latent space, and applying it for retrieval or steering. For a visualization of the approach, see Figure 1. See Figure 2 for an example.

**1. Creating a Concept Query**    The *Concept Query* $q_c \in \mathbb{R}^{d_c}$ is a vector that represents the concept of interest in terms of Concept Atlas features. We can construct it in several ways: Firstly, we can directly use the feature descriptions $\mathcal{D}$ of the Concept Atlas and set the indices of features that are relevant to the concept to a value of one, while setting the rest to zero. Secondly, we can use embedding models to embed both a human query and the feature descriptions $\mathcal{D}$ and retrieve the descriptions that are closest to it. We set the indices of relevant features to one (or a similarity score), and the rest to zero. Finally, similar to how one would create a steering vector, we can also build averaged or contrastive concept queries in the Concept Atlas, by forwarding a set of concept related sequences through the foundation model and aggregating their latent space activations. This flexibility allows us to use both human-guided and data-driven concept definitions.

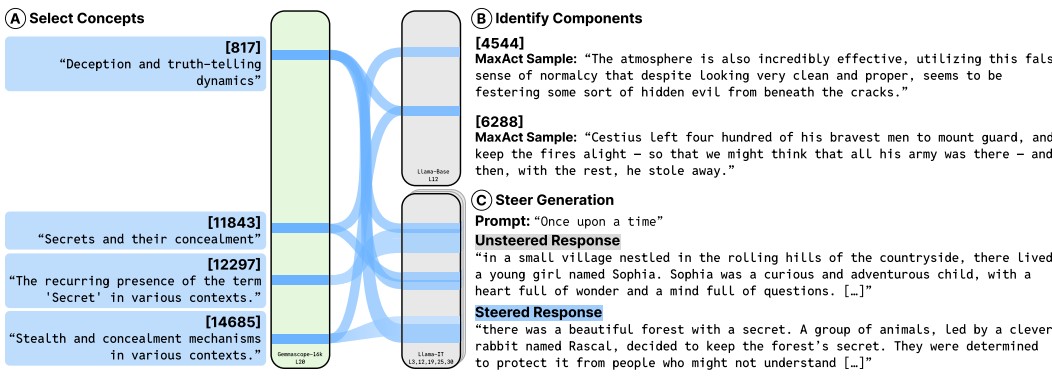

Figure 2: Examples of using **Atlas-Alignment** for identification and steering. (A) A Concept Query is constructed from multiple Concept Atlas features related to the theme of "secrets and deception" and mapped into the latent spaces of two subject models. (B) In Llama-Base, the alignment reveals two features in layer 12 that encode relevant concepts. (C) In Llama-IT, the same Concept Query is used to steer generation across multiple layers, shifting outputs toward concept-related text.

**2. Mapping to the subject model**  We map the concept query vector to the subject model space using the row-normalized cosine similarity between the query vector $q_c$ and the $\mathbf{T}_{s \to c}$ matrix

$$s_c = \widehat{\mathbf{T}}_{s \to c} \frac{q_c}{\|q_c\|_2} \quad \text{where} \quad \widehat{\mathbf{T}}_{s \to c \ [k,:]} = \frac{\mathbf{T}_{s \to c \ [k,:]}}{\|\mathbf{T}_{s \to c \ [k,:]}\|_2} \tag{3}$$

The resulting similarity vector $s_c \in \mathbb{R}^{d_s}$ scores each subject feature by its alignment with the chosen concept.

**3. Identification and Steering**  We use the similarity vector $s_c$ to *identify* which subject model features encode the target concept. Each entry of $s_c$ measures how strongly a feature aligns with the concept. Investigating the top-scoring features answers the question: "where, and to what degree, is the concept represented in the subject model's latent space?" In this way, the previously opaque feature space becomes searchable and interpretable.

The same vector $s_c$ also provides a direction for intervention. By adding a scaled version of $s_c$ to the subject model activations at inference time

$$a^{(\text{modified})} = \left( a^{(\text{original})} + \lambda\, s_c \right) \cdot \frac{\|a^{(\text{original})}\|_2}{\|a^{(\text{original})} + \lambda\, s_c\|_2} \tag{4}$$

we can steer the model's behavior along the chosen concept, analogous to steering with vectors derived from labeled datasets. Here $\lambda$ is a scalar that controls the strength of the intervention. Because $s_c$ is obtained without supervision, this provides a fast, concept-data free mechanism to control model behavior.

# 4    EXPERIMENTAL RESULTS

In section 4.1 we outline the implementation details. In section 4.2 we demonstrate how **Atlas-Alignment** can be used to identify relevant features and modify model outputs. In section 4.3 we quantitatively evaluate the semantic content transfer between Concept Atlas and subject models and in section 4.4 we quantitatively evaluate concept steering capabilities.

## 4.1    IMPLEMENTATION DETAILS

We use a subset of the Pile dataset (Gao et al., 2020), preprocessed as described in Appendix A.2, resulting in 1 million sequences and more than 30 million tokens for the Llama 3.1 family models. We evaluate five translation methods: covariance, correlation, linear regression, Orthogonal Procrustes with row-wise $L_2$-normalization, and Semantic Lens. Each method is trained on 500k samples, while a disjoint set of 500k samples is reserved for the qualitative and quantitative evaluation.

**Feature Identification**

**Concept Atlas Feature 13419:** "Chess-related terminology and figures."

| Llama-Base | L3;F4693 | White goes for a tactical blow19.Bxh7+ Kxh7 20.Rxd5; White is threatening to play Qe4+ followed by Rh5. |
| Llama-IT | L25;F10656 | Houdini 3 Pro will now support hash tables up to 256 GB.The engine evaluations have been carefully recalibrated so that +1.00 pawn advantage [..] |
| Llama-R1 | L19;F4561 | although an invasion on the seventh should still give him promising play in exchange for the material deficit [18.Rxc3 Ne4 19.Bxe7 Nxc3 [..] |

**Concept Atlas Feature 5095:** "Dream-related phenomena and vivid experiences."

| Llama-Base | L3;F13065 | But, our dreams often feel real to us despite blundering through the telling of them. |
| Llama-IT | L19;F12796 | In the car.' There wasn't anyone else in the car, and I decided that this was a hallucination, the result of tiredness and [..] |
| Llama-R1 | L25;F8319 | I managed to drift off to sleep again, and was jolted awake by the plane touching down in Charlotte. |

Table 1: Example of feature identification using Concept Atlas queries, mapped via Orthogonal Procrustes. The most similar feature and its maximally activating test sample are shown.

We use subject model latent spaces from four model families: GPT-2 (Radford et al., 2019) at layer 8; Qwen 2.5 Yang et al. (2024); Team (2024) (referred to here as Qwen) in the 0.5B configuration at layer 15, 1.5B at layer 17, and 3B at layer 21; Ministral-8B-Instruct-2410 (AI, 2024) (referred to here as Ministral) at layer 21; and the base and instruction-tuned variants of Llama 3.1 8B (Dubey et al., 2024), together with the R1-distilled version of Llama 3.1 8B Instruct (Guo et al., 2025) (referred to here as Llama-Base, Llama-IT, and Llama-R1) at layers 3, 12, 19, 25, and 30. For all models, we use features extracted from the MLP layers.

As Concept Atlases, we employ the Gemma 2 2B model (Riviere et al., 2024) with the Gemma Scope SAE encoder head at layer 20, in both the 16k (`average_l0_71`) and 65k (`average_l0_114`) configurations (Lieberum et al., 2024), trained on the residual stream activations and denoted Gemma Scope 16k and Gemma Scope 65k. Unless otherwise specified, we use labels for the Concept Atlas generated in Puri et al. (2025) using automated interpretability methods.

### 4.2 RESULTS

#### 4.2.1 IDENTIFICATION OF FEATURES

We use qualitative examples to show how the latent space translations can help us identify subject model features by linking them to queries from our Concept Atlas. Given a Concept Query, we compute the subject model feature similarities using the $\mathbf{T}_{s \to c}$ matrix. We then select the features with the highest scores and examine their maximally activating samples.

Table 1 shows representative examples of Concept Atlas features from Gemma Scope 16k mapped to Llama-Base, Llama-IT and Llama-R1 using the Orthogonal Procrustes method. Features with a high similarity show highest activations on samples that are semantically close to the queried concepts. This illustrates how a simple alignment via **Atlas-Alignment** makes it possible to identify concept-relevant features across different subject models.

#### 4.2.2 CONCEPT STEERING

We next provide qualitative results to illustrate how **Atlas-Alignment** enables steering across language models — modifying subject model outputs along concept directions.

Steering vectors are constructed from translated Concept Queries and added to the subject model activations, after which we examine the resulting generations. Steering with single directions often produces more incoherent generations, like endless repetitions of similar tokens. We instead combine several semantically related features to obtain more robust concept directions. This reflects the inherent difficulty of steering a model through its activations, given the non-linear connection between latent representations and output text.

In Table 2 we present examples of steering with Concept Queries from Gemma Scope 16k mapped to Llama-IT using Orthogonal Procrustes. We use a short seed prompt, let the model generate 100

**Concept Steering**

Prompt: "Sure, I always have a joke ready! Here is one:"

| | |
|---|---|
| Unmodified: | "Why did the scarecrow win an award? Because he was outstanding in his field. Get it? Outstanding in his field? Ahh, never mind. [..]" |
| Dogs and Cats: | "Why did the cat join a band? Because it wanted to be the purr-cussionist! Get it? Purr-cussionist? Like a percussionist, but with a cat [..]" |
| Reddit Comments: | "Why comment on this post? I'm not sure I understand the joke. Is it a joke about a joke? I'm not sure I understand the joke. [..]" |
| London: | "Why did the Londoner bring a ladder to the party? Because they wanted to take things to a higher level! (get it?) [..]" |

Table 2: Steering Llama-IT generation using Gemma Scope 16k Concept Atlas features translated with Orthogonal Procrustes.

tokens, and intervene with multiple steering factors $\lambda$. We apply steering simultaneously to layers 3, 12, 19, 25 and 30. Details on the used Concept Queries can be found in Appendix A.4. The examples show that steering with Concept Queries can meaningfully guide model outputs toward the chosen concept. Although steering is not always reliable, the results demonstrate that our framework enables subject models to inherit controllability from Concept Atlas features, turning them into semantically interpretable steering directions.

### 4.3 EVALUATING SEMANTIC TRANSLATIONS

Having qualitatively shown that **Atlas-Alignment** can transfer semantic concepts across models, we now turn to quantitative evaluation. Assessing translation quality is non-trivial, and we focus on two complementary goals. (1) General translation quality: how well does a translation align subject model latent spaces with the Concept Atlas? (2) Semantic retrieval: how useful is the translation for recovering semantic information from the subject model's latent space?

#### 4.3.1 TRANSLATION QUALITY

To measure how faithfully a translation connects the subject model latent space to the Concept Atlas, we evaluate how consistently it preserves feature-sample relationships. For a given Concept Atlas feature, we rank input samples by how strongly they activate that feature. We build a query vector containing only the atlas feature, translate it into the subject model space, and rank the samples based on the similarity to the concept vector $s_c$. Comparing the two rankings tells us whether the translation preserves the semantic signal.

We quantify this with the averaged AUROC and Average Precision (AP) metrics. AUROC captures how well the translated feature distinguishes samples that activate the Concept Atlas feature from those that do not, while AP emphasizes precision at the top of the ranking, measuring the degree to which the most salient concept samples remain highly ranked after translation.

We use the test split of 500k sequences and report averages for 100 randomly selected features from the Gemma Scope 16k Concept Atlas. For details on the sampled features see Appendix A.6.1.

Table 3 shows results for GPT-2, Qwen variants and Ministral, as well as multiple layers of Llama-Base. Orthogonal Procrustes consistently achieves the strongest performance across both AUROC and AP. The difference is most pronounced in AP, where Orthogonal Procrustes reaches values of 0.36-0.49 compared to a random baseline of 0.046. Inter-model differences appear to be similar in size to the intra-model scores of the Llama-Base layers. The relative performance of the different alignment methods largely transfers over the latent spaces. Similar patterns are observed for Llama-IT and Llama-R1 (see Appendix A.5).

To test the sensitivity of the translation quality on the training dataset size, we subsample the 500k train samples (roughly equivalent to 15M tokens) into subsets of 1k, 10k, 50k, 100k and 250k and create translation matrices for the Llama-IT layer 19 subject model and the Gemma-Scope 16k Concept Atlas. We then evaluate the alignment performance on the full 500k-sample test set. Results are shown in Figure 3 (left side). Orthogonal Procrustes dominates all other methods from 50k onward, with monotonic gains. Linear Regression peaks at 100k before degrading. Interestingly, even at 50k, Orthogonal Procrustes retains most of its performance and is stronger than all other

| Model | GPT-2 | | Qwen 0.5B | | Qwen 1.5B | | Qwen 3B | | Ministral | |
|---|---|---|---|---|---|---|---|---|---|---|
| **Method** | AUROC | AP | AUROC | AP | AUROC | AP | AUROC | AP | AUROC | AP |
| Linear Regression | 0.75 | 0.15 | 0.81 | 0.23 | 0.73 | 0.14 | 0.76 | 0.16 | 0.75 | 0.16 |
| Covariance | 0.79 | 0.19 | 0.82 | 0.23 | 0.80 | 0.20 | 0.82 | 0.23 | 0.82 | 0.24 |
| Cross Correlation | 0.78 | 0.19 | 0.82 | 0.23 | 0.80 | 0.20 | 0.82 | 0.23 | 0.82 | 0.24 |
| Semantic Lens | 0.72 | 0.14 | 0.77 | 0.18 | 0.73 | 0.14 | 0.76 | 0.16 | 0.78 | 0.19 |
| Orthogonal Procrustes | **0.82** | **0.36** | **0.84** | **0.39** | **0.84** | **0.41** | **0.83** | **0.42** | **0.86** | **0.49** |

| **Llama-Base** | Layer 3 | | Layer 12 | | Layer 19 | | Layer 25 | | Layer 30 | |
|---|---|---|---|---|---|---|---|---|---|---|
| **Method** | AUROC | AP | AUROC | AP | AUROC | AP | AUROC | AP | AUROC | AP |
| Linear Regression | 0.75 | 0.15 | 0.75 | 0.15 | 0.77 | 0.17 | 0.74 | 0.14 | 0.74 | 0.14 |
| Covariance | 0.81 | 0.23 | 0.79 | 0.19 | 0.82 | 0.24 | 0.78 | 0.18 | 0.78 | 0.16 |
| Cross Correlation | 0.81 | 0.22 | 0.79 | 0.19 | 0.82 | 0.24 | 0.78 | 0.18 | 0.78 | 0.16 |
| Semantic Lens | 0.77 | 0.19 | 0.73 | 0.13 | 0.78 | 0.19 | 0.74 | 0.14 | 0.71 | 0.11 |
| Orthogonal Procrustes | **0.83** | **0.43** | **0.82** | **0.39** | **0.86** | **0.49** | **0.83** | **0.44** | **0.83** | **0.40** |

Table 3: Translation quality measured in AUROC and AP for single-feature queries from the Gemma Scope 16k Concept Atlas. **Top:** Results for latent spaces of non-Llama Models. **Bottom:** Results for the Llama-Base model at different layers. Across all latent spaces Orthogonal Procrustes achieves the strongest results. For all results see Appendix A.5.

tested methods across all subsets, showing the efficacy of the method and the possibility of cutting compute cost without strong declines in quality. Full results are shown in Table 8 in the Appendix.

We also examine the impact of the pooling function used to aggregate token-level activations (see Section 3.1), comparing max- to mean-pooling. We construct mean-pooled translation matrices for the subject models GPT-2, Qwen 3B, and Llama-IT layer 19 and the Gemma Scope 16k Concept Atlas, and evaluate them against the max-pooled variants. Results are mixed and method-dependent. Covariance, Cross Correlation, and Semantic Lens generally improve with mean-pooling, yielding higher AP scores across all latent spaces and slight AUROC gains in GPT-2 and Qwen 3B. In contrast, Linear Regression and Orthogonal Procrustes degrade substantially under mean-pooling. Orthogonal Procrustes remains the strongest method overall, retaining the highest AP in both pooling variants while max-pooling further gives it the best AUROC score across all configurations. This shows that pooling choice can strongly influence alignment quality and should be considered jointly with the translation method selection. Full results are in Table 9 in the Appendix.

### 4.3.2 SEMANTIC RETRIEVAL

We next evaluate how well translations enable the retrieval of specific semantic features from the subject model. This is particularly relevant for identifying subject features tied to concrete concepts.

To test this, we use ground-truth feature annotations. For each subject model feature $i$, we generate a set of concept-related input sequences $X^{(i)}$ that reliably activate it. Averaging their max-pooled activations in the Concept Atlas yields a targeted Concept Query $q_c^{(i)}$. We map this query into the subject model space via $\mathbf{T}_{s \to c}$, and obtain similarity vector $s_c^{(i)} \in \mathbb{R}^{d_s}$.

We evaluate retrieval performance using two ranking-based metrics: The Mean Reciprocal Rank (MRR) reflects how highly the correct feature is ranked, with $1$ indicating perfect retrieval and $1/d_s$ corresponding to random guessing. The Mean Predicted Probability (MPP) captures the confidence in the choice, by assigning a softmax-normalized probability to the correct feature after z-score normalization of the similarity. For formal definitions see Appendix A.6.

We use subject model features from Llama-IT Layer 19 and feature descriptions generated in Choi et al. (2024). We generate 20 synthetic input sequences per feature to form the ground-truth concept sets, on the basis of which we retrieve the features. To ensure validity, we (i) only include features with reliable descriptions, here defined as features with a harmonic mean score $> 0.75$ on the activation-based FADE metrics (Puri et al., 2025) and (ii) discard features where $X^{(i)}$ does not maximally activate the chosen feature in the subject model latent space. This leaves us with a set of 454 validated features. Results are reported for Gemma Scope 16k and 65k Concept Atlases.

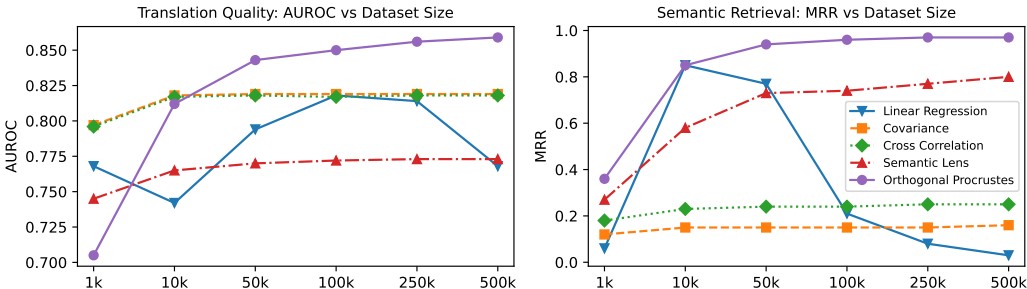

Figure 3: Alignment performance vs training subset scale (1k–500k), evaluated on the full 500k-sample test set. **Left:** Translation quality, reported in AUROC. **Right:** Semantic retrieval quality, reported in MRR. Full results are in the Appendix (Tables 8 and 10).

|  | Gemma Scope 16k | | | | Gemma Scope 65k | | | |
|  | MRR | | MPP | | MRR | | MPP | |
| **Method** | mean | std | mean | std | mean | std | mean | std |
| Linear Regression | 0.03 | 0.11 | 0.00 | 0.00 | 0.02 | 0.06 | 0.00 | 0.00 |
| Covariance | 0.16 | 0.30 | 0.01 | 0.03 | 0.12 | 0.27 | 0.01 | 0.02 |
| Cross Correlation | 0.25 | 0.36 | 0.01 | 0.04 | 0.26 | 0.37 | 0.02 | 0.04 |
| Semantic Lens | 0.80 | 0.36 | 0.05 | 0.12 | 0.79 | 0.37 | 0.05 | 0.13 |
| Orthogonal Procrustes | **0.97** | 0.13 | **0.92** | 0.21 | **0.97** | 0.12 | **0.94** | 0.19 |

Table 4: Semantic retrieval from Llama-IT Layer 19 into Gemma Scope 16k and 65k, measured with MRR and MPP. Random baseline scores: $MRR = 0.0147$, $MPP = 0.0022$. Orthogonal Procrustes consistently outperforms all other methods.

Table 4 shows that Orthogonal Procrustes achieves near-perfect retrieval, with MRR and MPP indicating the correct feature is recovered almost every time and with high confidence. Linear regression, covariance, and cross-correlation yield much lower scores, with linear regression close to random baselines. Semantic Lens performs well in MRR but falls behind Orthogonal Procrustes on MPP. The retrieval evaluation shows significant differences between the various translation methods, highlighting the importance of the choice. It also shows that we can reliably retrieve relevant features using the Orthogonal Procrustes translation method.

To measure the sensitivity of the semantic retrieval on dataset size we again perform the training-data ablation experiment, where the representational alignment methods are used on progressively larger subsampled subsets and tested on the full 500k-sample set.

Results are shown in Figure 3 (right side). Orthogonal Procrustes again leads from 50k onward, improving with each larger subset. Linear Regression performs well at the 10k subset, but deteriorates strongly for more samples. Semantic Lens increases monotonically with more training data and Covariance and Cross-Correlation plateau early and stay flat across all subset sizes. Full results are in Table 10 in the Appendix.

## 4.4 EVALUATING STEERING

Finally, we evaluate how well different translation methods enable steering a subject model toward specific concepts using only Concept Atlas features. For an evaluation of cross-model supervised steering vectors, we refer to Huang et al. (2025).

We construct queries from multiple semantically related Concept Atlas features and map them into the subject model's latent space. The resulting vectors are injected as steering directions, and we measure how strongly they increase the expression of the targeted concept in the generated outputs. We quantify steering effectiveness using LLM-based ratings: generated sequences are classified as expressing the target concept class or not. Faithfulness is defined as the maximal relative increase in

| | Llama-Base | | | | | | | | | | | |
|---|---|---|---|---|---|---|---|---|---|---|---|---|
| Method | L3 | | L12 | | L19 | | L25 | | L30 | | Combined | |
| | $f$ | $\Delta a$ | $f$ | $\Delta a$ | $f$ | $\Delta a$ | $f$ | $\Delta a$ | $f$ | $\Delta a$ | $f$ | $\Delta a$ |
| Random | 2.92 | −0.27 | 3.20 | −0.14 | 3.96 | 0.29 | 2.44 | −0.26 | 4.01 | −0.04 | 1.88 | 0.88 |
| Cov | 3.39 | 0.27 | 4.63 | 0.22 | 4.60 | 0.21 | **4.40** | −0.05 | 11.39 | 1.89 | 13.02 | 1.44 |
| CrossCorr | 4.46 | −0.07 | **5.04** | 0.48 | 6.07 | 0.51 | 3.33 | 0.10 | 11.70 | 1.62 | 8.26 | 2.35 |
| LinReg | 3.40 | −0.03 | 4.86 | 0.38 | 4.25 | −0.37 | 3.39 | 0.06 | 4.28 | −0.18 | 5.05 | −0.32 |
| SemLen | 5.32 | −0.29 | 3.61 | −0.30 | 3.18 | −0.52 | 2.50 | 0.41 | 10.17 | 1.40 | 9.70 | 0.93 |
| OrthProc | **6.77** | 0.10 | 4.24 | −0.14 | **8.96** | 0.65 | 4.17 | 0.04 | **32.62** | 4.32 | **31.42** | 3.83 |
| | Llama-IT | | | | | | | | | | | |
| Random | 2.94 | 0.40 | 5.00 | 0.68 | 4.97 | −0.02 | 3.27 | 0.37 | 3.15 | 0.06 | 2.72 | −0.01 |
| Cov | 3.76 | 0.29 | 4.97 | 0.33 | **6.97** | 0.59 | 3.53 | 0.38 | 12.08 | 3.88 | 12.21 | 1.81 |
| CrossCorr | 3.81 | 0.49 | 5.40 | 0.51 | 4.85 | 0.54 | 4.79 | 0.39 | 10.22 | 2.68 | 11.26 | 1.40 |
| LinReg | 4.46 | 0.28 | 5.58 | 1.13 | 6.68 | 0.44 | 3.30 | −0.04 | 3.69 | 0.07 | 2.79 | −0.82 |
| SemLen | **7.50** | 0.23 | 5.88 | 0.66 | 5.70 | 0.46 | **5.80** | 0.19 | 8.97 | 1.70 | 10.65 | 2.02 |
| OrthProc | 6.59 | 0.36 | **6.44** | 0.98 | 4.98 | 0.20 | 5.67 | 0.33 | **30.98** | 6.04 | **41.80** | 6.66 |

Table 5: Steering results on Llama-Base and Llama-IT measured with faithfulness ($f$) and mean activation change ($\Delta a$). Random steering provides the baseline. Steering up to layer 25 shows smaller effects, while layer 30 shows strong increases, especially for Orthogonal Procrustes.

concept expression over the no-steering baseline,

$$\text{faithfulness} = \max_i \frac{r_{\lambda_i} - r_{\lambda_0}}{1 - r_{\lambda_0}} \times 100 \tag{5}$$

where $r_{\lambda_i}$ is the share of concept-related generations at steering factor $\lambda_i$, and $r_{\lambda_0}$ is the baseline share without steering. We report the faithfulness score as a percentage, averaged across Concept Queries. As a complementary metric, we report the mean change in the Concept Atlas feature activations. Here we exclude layers or queries where steering has no effects.

We construct 10 Concept Queries from Gemma Scope 16k and apply them to Llama-Base and Llama-IT. Each query is evaluated on 16 seed prompts with 100-token continuations. Steering factors are $\lambda \in [-50, -10, -1, 0, 1, 10, 50]$. As a baseline, we use random steering directions. Outputs are rated by `gpt-4o-mini-2024-07-18` (OpenAI, 2024), sampled three times with temperature 1, using the median label. Further details are provided in Appendix A.7.

Table 5 shows that steering in early and mid layers (3–25) results only in small gains over the random baseline, with faithfulness rarely above 7–9%. In contrast, layer 30 exhibits pronounced effects: covariance, cross-correlation, and Semantic Lens reach 9–12%, while Orthogonal Procrustes exceeds 30% on Llama-Base and 40% on Llama-IT, with strong activation changes.

These results indicate that effective atlas-based steering is concentrated in the final layers, and that Orthogonal Procrustes provides the most reliable path to semantically controllable interventions.

## 5 DISCUSSION

**Limitations:** While we demonstrate reliable transfer of semantic concepts between the Gemma Scope Concept Atlas and the Llama 3.1 family models, our framework relies on the assumption that different models learn comparable concepts, as suggested by the Platonic and Linear Representation Hypotheses. Similar to the training of SAEs, it also requires that the translation dataset contains sufficient variety to cover the full set of atlas features and concepts. In addition, our current design discards positional information through the max-pooling of activations. Future work could address these limitations and further test the robustness of the framework.

**Conclusion:** In this work, we introduce **Atlas-Alignment**, a framework for transferring interpretability across language models by aligning unknown latent spaces to a well-understood Concept Atlas. We show that lightweight alignment methods, particularly Orthogonal Procrustes, enable robust semantic transfer, reliable concept retrieval, and controllable generation based solely on Concept Atlas features. We hope that this approach can serve as a starting point for future investigations into cross-model alignment as a foundation for interpretability, and encourages the development and evaluation of further alignment methods. More broadly, we hope to make interpretability more scalable and effective by allowing a single high-quality atlas to be used across many different models.

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

# A  APPENDIX

## A.1  LICENSES

`Gemma-2-2b` is released under a custom Gemma Terms of Use. `Gemma Scope` SAEs are released under Creative Commons Attribution 4.0 International. `Llama3.1-8B`, `Llama3.1-8B-Instruct` are released under a custom Llama 3.1 Community License. Pile Uncopyrighted dataset and `Deepseek R1-Distill-Llama-8B` is released under MIT License.

## A.2  DATASET GENERATION

For our experiments, we use a subset of the uncopyrighted version of the Pile dataset (Gao et al., 2020), with all copyrighted content removed. Following the procedure outlined in Puri et al. (2025), we sample from the test partition while preserving the relative proportions of the original data sources. The sampled texts are further preprocessed using the NLTK sentence tokenizer (Bird et al., 2009) to divide larger passages into smaller sequences. We then filter out sentences in the bottom and top fifth percentiles of length, which typically corresponded to out-of-distribution cases such as single words, isolated characters, or unusually long outliers. Next, we remove sentences consisting only of numbers or special characters and deduplicate the resulting set.

The final dataset contains 1M sequences, averaging 120.4 characters (SD 72.5) with lengths ranging from 2 to 391. Using the Llama 3.1 8B tokenizer, these correspond to an average of 30.1 tokens (SD 18.6). We split the data into two subsets of 500k samples each for training and evaluation.

## A.3  TRANSLATION METHODS

We list here further translation methods used in this work:

- **Covariance:**

$$\text{translate}_{\text{Cov}}(A_s, A_c) = \tilde{A}_s^\top \tilde{A}_c \quad \text{with} \quad \tilde{A}_i = A_i - \mu_i \tag{6}$$

  where $\mu_i$ is the vector of column means of $A_i$.

- **Correlation:**

$$\text{translate}_{\text{Corr}}(A_s, A_c) = D_s^{-1} \tilde{A}_s^\top \tilde{A}_c D_c^{-1} \quad \text{with} \quad \tilde{A}_i = A_i - \mu_i, \ D_i = \text{diag}(\sigma_i), \tag{7}$$

  where $\mu_i$ is the vector of column means of $A_i$ and $\sigma_i$ is the vector of column standard deviations of $A_i$.

- **Linear Regression:** A straightforward way to translate feature spaces is by using linear regression. It finds the translation that minimizes squared reconstruction error between subject and atlas activations

$$\text{translate}_{\text{OLS}}(A_s, A_c) = \underset{\mathbf{T}_{s \to c}}{\arg\min} \|A_s - A_c \mathbf{T}_{s \to c}^\top\|_F^2 \tag{8}$$

- **Semantic Lens:** A simplified version of Semantic Lens (Dreyer et al., 2025) can similarly be applied. It represents each subject model feature by the subset of most activating samples. Specifically, we keep the top-k activations per feature, binarize them, and average the corresponding Concept Atlas embeddings

$$\text{translate}_{\text{SL}}(A_s, A_c) = \tilde{A}_s^\top A_c \quad \text{with} \quad \tilde{A}_{s\,[ij]} = \frac{1}{k} \cdot \mathbf{1}\left\{A_{s\,[ij]} \in \text{TopK}(A_{s\,[i,:]}, k)\right\} \tag{9}$$

  where TopK returns the indices of the largest $k$ values for the feature column $A_{s[i,:]}$, thus selecting the most salient samples per subject model feature. This describes each subject feature as the mean Concept Atlas embedding of its most salient samples.

## A.4  QUALITATIVE STEERING QUERIES

We use queries generated from Concept Atlas features from the Gemma Scope 16k SAE in layer 20. All features are weighted equally with a weight of 1. In table 6 we name the concept and the feature numbers along with the used modification factor.

| Concept | Features | Modification Factor |
|---|---|---|
| reddit comments | [1786, 13945, 9829, 9346, 9736, 13851, 7937, 1914, 2402, 3204, 12203, 10075, 1917, 5067] | 50 |
| dogs and cats | [6772, 1089, 12082, 13747] | 50 |
| london | [5218, 12614] | 35 |

Table 6: Concepts and their corresponding features from the Gemma Scope 16k SAE used in the qualitative steering examples.

| | LinReg | | Cov | | CrossCorr | | SemLen | | OrthProc | |
|---|---|---|---|---|---|---|---|---|---|---|
| | AUROC | AP | AUROC | AP | AUROC | AP | AUROC | AP | AUROC | AP |
| **GPT-2** | 0.75 | 0.15 | 0.79 | 0.19 | 0.78 | 0.19 | 0.72 | 0.14 | **0.82** | **0.36** |
| **Qwen 0.5B** | 0.81 | 0.23 | 0.82 | 0.23 | 0.82 | 0.23 | 0.77 | 0.18 | **0.84** | **0.39** |
| **Qwen 1.5B** | 0.73 | 0.14 | 0.80 | 0.20 | 0.80 | 0.20 | 0.73 | 0.14 | **0.84** | **0.41** |
| **Qwen 3B** | 0.76 | 0.16 | 0.82 | 0.23 | 0.82 | 0.23 | 0.76 | 0.16 | **0.83** | **0.42** |
| **Ministral** | 0.75 | 0.16 | 0.82 | 0.24 | 0.82 | 0.24 | 0.78 | 0.19 | **0.86** | **0.49** |
| **Llama-Base** | | | | | | | | | | |
| L3 | 0.75 | 0.15 | 0.81 | 0.23 | 0.81 | 0.22 | 0.77 | 0.19 | **0.83** | **0.43** |
| L12 | 0.75 | 0.15 | 0.79 | 0.19 | 0.79 | 0.19 | 0.73 | 0.13 | **0.82** | **0.39** |
| L19 | 0.77 | 0.17 | 0.82 | 0.24 | 0.82 | 0.24 | 0.78 | 0.19 | **0.86** | **0.49** |
| L25 | 0.74 | 0.14 | 0.78 | 0.18 | 0.78 | 0.18 | 0.74 | 0.14 | **0.83** | **0.44** |
| L30 | 0.74 | 0.14 | 0.78 | 0.16 | 0.78 | 0.16 | 0.71 | 0.11 | **0.83** | **0.40** |
| **Llama-IT** | | | | | | | | | | |
| L3 | 0.75 | 0.15 | 0.81 | 0.22 | 0.81 | 0.22 | 0.77 | 0.18 | **0.83** | **0.43** |
| L12 | 0.74 | 0.15 | 0.79 | 0.19 | 0.79 | 0.19 | 0.72 | 0.13 | **0.81** | **0.38** |
| L19 | 0.77 | 0.17 | 0.82 | 0.23 | 0.82 | 0.23 | 0.77 | 0.18 | **0.86** | **0.49** |
| L25 | 0.74 | 0.14 | 0.78 | 0.18 | 0.78 | 0.18 | 0.73 | 0.14 | **0.83** | **0.44** |
| L30 | 0.73 | 0.13 | 0.77 | 0.16 | 0.77 | 0.16 | 0.70 | 0.11 | **0.83** | **0.39** |
| **Llama-R1** | | | | | | | | | | |
| L3 | 0.73 | 0.14 | 0.80 | 0.21 | 0.79 | 0.21 | 0.76 | 0.17 | **0.82** | **0.40** |
| L12 | 0.73 | 0.13 | 0.78 | 0.17 | 0.77 | 0.17 | 0.71 | 0.12 | **0.80** | **0.36** |
| L19 | 0.75 | 0.16 | 0.80 | 0.22 | 0.80 | 0.22 | 0.75 | 0.17 | **0.84** | **0.47** |
| L25 | 0.72 | 0.13 | 0.76 | 0.17 | 0.76 | 0.17 | 0.72 | 0.13 | **0.82** | **0.42** |
| L30 | 0.73 | 0.13 | 0.76 | 0.15 | 0.76 | 0.15 | 0.70 | 0.10 | **0.81** | **0.37** |

Table 7: Full evaluation of single-feature queries from the Gemma Scope 16k Concept Atlas across GPT-2, Qwen 0.5B, Qwen 1.5B and Qwen 3B, Ministral as well as the Llama-Base, Llama-IT, and Llama-R1 models at different layers. Reported are AUROC and AP. Orthogonal Procrustes consistently yields the highest scores across all latent spaces.

## A.5 EVALUATION OF TRANSLATION QUALITY

Full Results for models Llama-Base, Llama-IT and Llama-R1 are presented in Table 7.

### A.5.1 TRANSLATION QUALITY DATASET ABLATION

Results of the dataset-ablation experiment for Llama-IT layer 19 to Concept Atlas Gemma Scope 16k are presented in Table 8.

| | LinReg | | Cov | | CrossCorr | | SemLen | | OrthProc | |
|---|---|---|---|---|---|---|---|---|---|---|
| | AUROC | AP | AUROC | AP | AUROC | AP | AUROC | AP | AUROC | AP |
| **1k** | 0.77 | 0.19 | 0.80 | 0.22 | 0.80 | 0.22 | 0.75 | 0.15 | 0.71 | 0.21 |
| **10k** | 0.74 | 0.25 | 0.82 | 0.23 | 0.82 | 0.23 | 0.77 | 0.17 | 0.81 | 0.39 |
| **50k** | 0.79 | 0.32 | 0.82 | 0.24 | 0.82 | 0.23 | 0.77 | 0.18 | 0.84 | 0.46 |
| **100k** | 0.82 | 0.28 | 0.82 | 0.24 | 0.82 | 0.23 | 0.77 | 0.18 | 0.85 | 0.48 |
| **250k** | 0.81 | 0.24 | 0.82 | 0.24 | 0.82 | 0.23 | 0.77 | 0.18 | 0.86 | 0.49 |
| **500k** | 0.77 | 0.17 | 0.82 | 0.24 | 0.82 | 0.23 | 0.77 | 0.18 | 0.86 | 0.49 |

Table 8: Translation Quality from Llama-IT Layer 19 to Concept Atlas Gemma Scope 16k at different subsets of the train dataset.

### A.5.2 TRANSLATION QUALITY POOLING CHOICE EXPERIMENT

Results of the pooling comparison experiment for GPT-2, Qwen 3B and Llama-IT layer 19 to Concept Atlas Gemma Scope 16k are presented in Table 9.

| Model | Pooling | LinReg | | Cov | | CrossCorr | | SemLen | | OrthProc | |
|---|---|---|---|---|---|---|---|---|---|---|---|
| | | AUROC | AP | AUROC | AP | AUROC | AP | AUROC | AP | AUROC | AP |
| GPT-2 | Mean | 0.70 | 0.13 | 0.80 | 0.23 | 0.80 | 0.23 | 0.77 | 0.22 | 0.79 | 0.31 |
| GPT-2 | Max | 0.75 | 0.15 | 0.79 | 0.19 | 0.78 | 0.19 | 0.72 | 0.14 | 0.82 | 0.36 |
| Qwen 3B | Mean | 0.73 | 0.15 | 0.81 | 0.25 | 0.81 | 0.25 | 0.79 | 0.26 | 0.81 | 0.35 |
| Qwen 3B | Max | 0.76 | 0.16 | 0.82 | 0.23 | 0.82 | 0.23 | 0.76 | 0.16 | 0.83 | 0.42 |
| Llama-IT (Layer 19) | Mean | 0.71 | 0.15 | 0.81 | 0.27 | 0.81 | 0.27 | 0.76 | 0.24 | 0.84 | 0.44 |
| Llama-IT (Layer 19) | Max | 0.77 | 0.17 | 0.82 | 0.24 | 0.82 | 0.23 | 0.77 | 0.18 | 0.86 | 0.49 |

Table 9: Comparison of Max- and Mean-Pooling Strategies for the models GPT-2, Qwen 3b and Llama-IT layer 19 and the Gemma Scope 16k Concept Atlas measured using the translation quality evaluation.

### A.6 SEMANTIC RETRIEVAL

The Mean Reciprocal Rank (MRR) measures how highly the correct feature is ranked relative to all others:

$$\text{MRR} = \frac{1}{d_s} \sum_{i=1}^{d_s} \frac{1}{1 + \sum_{k \neq i} \mathbf{1}\left[s_i^{(i)} < s_k^{(i)}\right]}. \tag{10}$$

A score of 1 indicates perfect retrieval, while $1/d_s$ corresponds to random guessing.

The Mean Predicted Probability (MPP) measures the softmax-normalized probability assigned to the correct feature after z-score normalization:

$$\text{MPP} = \frac{1}{d_s} \sum_{i=1}^{d_s} \frac{\exp\left(\tilde{s}_i^{(i)}\right)}{\sum_{k=1}^{d_s} \exp\left(\tilde{s}_k^{(i)}\right)}, \tag{11}$$

where $\tilde{s}^{(i)}$ is the similarity vector standardized by z-score normalization.

### A.6.1 IDENTIFICATION QUERIES

We use the following 100 Concept Atlas features from the Gemma Scope 16k L20 SAE to evaluate the strength of ranking capabilities:

```
[464, 470, 496, 648, 662, 708, 775, 837, 908, 1029, 1031, 1217, 1287, 1375, 1554, 1555, 1768, 1796, 1814,
1837, 2385, 2423, 2483, 2712, 2717, 2720, 2782, 2985, 3052, 3258, 4928, 5086, 5219, 5271, 5485, 5544, 5908,
5986, 5992, 6226, 6270, 6371, 6419, 6441, 6525, 6770, 6902, 6930, 7082, 7107, 7190, 7215, 7230, 7291, 7384,
7414, 7647, 7877, 8030, 8332, 8346, 8377, 8391, 8438, 8489, 8598, 8779, 9453, 9622, 9680, 9703, 9743, 9785,
10158, 10242, 10428, 10793, 10819, 10964, 11044, 11200, 11318, 11463, 11668, 11769, 12116, 12373, 12417,
12516, 12549, 12744, 13215, 13293, 13437, 13547, 13708, 13719, 14131, 14660, 14694]
```

### A.6.2 SEMANTIC RETRIEVAL DATASET ABLATION

Results of the dataset-ablation experiment for Llama-IT layer 19 to Concept Atlas Gemma Scope 16k are presented in Table 10.

### A.7 STEERING EXPERIMENT

The following seed sequences are used to initialize model generations:

```
"Once upon a time",
"I just started",
"The candidate",
"Section 1",
"Documents",
```

| | LinReg | | Cov | | CrossCorr | | SemLen | | OrthProc | |
|---|---|---|---|---|---|---|---|---|---|---|
| | MRR | MPP | MRR | MPP | MRR | MPP | MRR | MPP | MRR | MPP |
| **1k** | 0.06 | 0.01 | 0.12 | 0.01 | 0.18 | 0.01 | 0.27 | 0.01 | 0.36 | 0.15 |
| **10k** | 0.85 | 0.54 | 0.15 | 0.01 | 0.23 | 0.01 | 0.58 | 0.03 | 0.85 | 0.68 |
| **50k** | 0.77 | 0.35 | 0.15 | 0.01 | 0.24 | 0.01 | 0.73 | 0.04 | 0.94 | 0.87 |
| **100k** | 0.21 | 0.03 | 0.15 | 0.01 | 0.24 | 0.01 | 0.74 | 0.04 | 0.96 | 0.89 |
| **250k** | 0.08 | 0.01 | 0.15 | 0.01 | 0.25 | 0.01 | 0.77 | 0.05 | 0.97 | 0.91 |
| **500k** | 0.03 | 0.00 | 0.16 | 0.01 | 0.25 | 0.01 | 0.80 | 0.36 | 0.97 | 0.92 |

Table 10: Semantic Retrieval from Llama-IT Layer 19 via Concept Atlas Gemma Scope 16k at different subsets of the train dataset.

```
"The capital of",
"Let me tell you a joke",
" ",
"French cuisine",
"do_sample=False",
"Would you like to know",
"It is widely used",
"\\scalebox{0.85}{$\\mathcal{L} =",
"The team struggled",
"Hallo, woher denn",
"When you say",
```

To automatically rate the outputs, we use the OpenAI model `gpt-4o-mini-2024-07-18` (OpenAI, 2024). The model is instructed to classify sequences according to whether a given concept is expressed: class 0 means the concept is absent, class 2 means it is clearly present, and class 1 (vague/partial) is excluded to reduce spurious correlations. The rating prompt is:

```
You are given a TOPIC and a set of sequences. Score each sequence for whether the TOPIC is expressed,
counting either literal string matches or clear semantic references (either is sufficient).
Scoring: 2 = present. 1 = vague/partial. 0 = not expressed. Score independently per sequence.
Each sequence has a unique ID. Provide your ratings as a Python dictionary with sequence IDs as keys and
respective ratings as values.
Output only the dictionary - no additional text, comments, or symbols.
Below we show examples of input and outputs for three topics.

Example Input: 'Concept: Germany\n
Sequence ID 0: "German scientists unveiled a new fusion experiment."\n
Sequence ID 1: "Berlin's Brandenburg Gate was illuminated last night."\n
Sequence ID 2: "A Munich-based automaker expanded its EV lineup."\n
Sequence ID 3: "Several European countries adjusted tax policy."\n
Sequence ID 4: "The federal republic that reunified in 1990 tightened export rules."\n
Sequence ID 5: "country = \"Deutschland\""\n
Sequence ID 6: "Mount Fuji is a popular destination for hikers."\n
Sequence ID 7: "Silicon Valley companies are racing ahead in AI."'
Example Output: '{{\"0\": 2, \"1\": 2, \"2\": 2, \"3\": 0, \"4\": 2, \"5\": 2, \"6\": 0, \"7\": 0}}'

Example Input: 'Concept: Football\n
Sequence ID 0: "The quarterback threw a 60-yard touchdown pass."\n
Sequence ID 1: "The UEFA Champions League final kicks off on Saturday."\n
Sequence ID 2: "topic = \"football\""\n
Sequence ID 3: "#football fans filled the stadium after the derby."\n
Sequence ID 4: "The offside rule was explained by the referee."\n
Sequence ID 5: "The match ended 2-1 after extra time."\n
Sequence ID 6: "The chef prepared sushi with fresh tuna."\n
Sequence ID 7: "Quantum entanglement was demonstrated in a new experiment."'
Example Output: '{{\"0\": 2, \"1\": 2, \"2\": 2, \"3\": 2, \"4\": 2, \"5\": 1, \"6\": 0, \"7\": 0}}'

Example Input: 'Concept: Paris\n
Sequence ID 0: ", there was a young Parisian named Hugo. He found a book filled with maps."\n
Sequence ID 1: "Paris, Texas, USA, 2005, 2006, 2007, 2008, 2009, 2010, 2011, 2012, 2013, 2014, 2015, 2016..."\n
Sequence ID 2: "renowned for its cuisine, and the Parisians are proud of their heritage."\n
Sequence ID 3: "# (1)\nimport numpy as np"\n
Sequence ID 4: "to use the `git` command to get the latest version of the `paris` tool, run these steps..."\n
Sequence ID 5: "You can write 'konnichiwa' in the title, but not the word 'Tokyo'."'
Example Output: '{{\"0\": 2, \"1\": 2, \"2\": 2, \"3\": 0, \"4\": 2, \"5\": 0}}'
```

### A.7.1 EVALUATION STEERING QUERIES

We use queries generated from multiple Concept Atlas features from the Gemma Scope 16k SAE in layer 20. All features are weighted equally with a weight of 1. In table 11 we show the broad concepts and the feature numbers.

## A.8 USE OF LARGE LANGUAGE MODELS

We used large language models to polish and refine the text for clarity and style.

| Concept | Features |
|---|---|
| reddit comments | [1786, 13945, 9829, 9346, 9736, 13851, 7937, 1914, 2402, 3204, 12203, 10075, 1917, 5067] |
| dreams and imagination | [5095, 6195, 320, 9017, 9273, 7225, 11922, 1974, 5755, 6576, 13207, 7342, 10331, 2104, 12727, 1631, 10669, 14509, 8630] |
| gardening | [6607, 568, 1689, 13279, 5514, 10459, 3138, 9328, 6056, 1676, 12871, 10010, 5680, 7747, 10759, 6369, 9839, 6316, 9125, 10678, 7360, 12587, 5317, 9396, 2725] |
| science fiction and fantasy | [6020, 9273, 8850, 8544, 3088, 6139, 3120, 6561, 2942, 11922, 2777, 8877, 5267, 6990, 2212, 11512, 13710, 1659, 7995, 10004, 779, 12857, 12899, 7365, 8927, 13805, 13602] |
| eating | [13834, 11544, 1351, 2793, 11867, 5898, 7683, 5531, 9027, 1247, 8513, 9750, 11847, 12394, 5838, 13497, 6621, 11491, 184, 8337, 8991, 668, 1538, 14334, 2480, 1632, 8771, 6657, 9125, 6847, 2247, 13567, 6643] |
| smart devices | [6211, 11318, 257, 5110, 8672, 7105, 14663, 12058, 8356, 1341, 13177, 13000, 948, 5950, 5078, 5791, 13434, 8092, 2942, 5833, 8450, 11350, 8124, 14292, 63, 13363, 1446] |
| driving and cars | [856, 6418, 11182, 3178, 6571, 5814, 11212, 11620, 5877, 14326, 7054, 7732, 8038, 10526, 6964, 10870, 14386, 11957, 10196, 11028, 14566, 13018, 11309, 2724, 9791, 3154, 6375, 7224, 1336, 10172, 207, 9032, 6767, 14137] |
| dogs and cats | [6772, 1089, 12082, 13747] |
| video games | [8877, 13641, 12839, 13962, 3016, 12805, 13317, 13596, 13064, 7522, 14643, 10390, 5864, 8026, 67, 3089, 5380, 2003, 211, 9270, 1537, 14751, 12039, 4950, 6538, 14259, 1276, 5979, 2942] |
| health and well-being | [12413, 986, 5090, 13997, 6624, 12235, 668, 2162, 10317, 155, 11583, 12425, 9404, 2963, 11867, 7943, 2009, 1705, 2912, 10078, 13216, 12593, 1462, 10355, 49, 11043, 5522, 8521] |

Table 11: Multi Query Features and the corresponding features from the Gemma Scope 16k SAE used in the steering evaluation.

