# OpenReview forum: "Atlas-Alignment: Making Interpretability Transferable Across Language Models"
_ICLR.cc/2026/Conference — Submitted to ICLR 2026_

### Official Review · Reviewer_5Awd · 2025-10-31

**Soundness:** 3
**Presentation:** 2
**Contribution:** 2
**Rating:** 2
**Confidence:** 4

**Summary:**

This paper proposes a framework, Atlas-Alignment, for transferring interpretable concept dictionaries, such as those learned by Sparse Autoencoders, across language models. They measure the utility of this framework for semantic search and model steering, finding that representational alignment across models allows for interpretability transferral.

**Strengths:**

- The application of the platonic representation hypothesis to interpretability is an interesting and active field of research right now.
- The authors test a variety of methods for cross-model alignment, including covariance, correlation, linear regression, Orthogonal Procrustes with row-wise L2-normalization, and SemanticLens.
- The authors present promising results for using Atlas-Alignment for cross model steering.

**Weaknesses:**

- The motivation for this work hinges almost entirely on the idea that existing concept discovery methods (e.g. SAEs) are very costly to use; however, it is my understanding that this is not necessarily true. Prior work has noted that SAEs can be trained in a couple of hours on a single GPU, and automated labelling can be done with open-source models. Is there any benefit to Atlas-Alignment outside of this cost? Furthermore, Atlas-Alignment still requires the use of an existing Concept Atlas, which is generally just a pretrained SAE. Are there any reasonable alternatives to this?
- The authors train their mappings with 500k samples, which seems equivalent to the amount of data required to train an SAE. Is it possible to scale this down to low data regimes?
- I also believe that the authors have overstated the difference between their work and existing literature that has argued for cross-model dictionaries or sparse autoencoders, such as [1, 2, 3] below. They simply note that previous "approaches require expensive training and can additionally suffer from high reconstruction loss" without providing empirical evidence for their training being less expensive, having higher reconstruction, or resulting in more effective steering. Providing quantitative comparisons with existing baselines across these axes would greatly strengthen this work.
- The authors do not compare their framework to any strong baselines for any of the experiments. If they are claiming that Atlas-Aligned SAEs allow for cross-model steering, then they should at least compare against steering with SAEs trained for the target model to evaluate for any loss in performance.


[1] Shared Global and Local Geometry of Language Model Embeddings. Andrew Lee, Melanie Weber, Fernanda Viegas, Martin Wattenberg.
[2] Universal Sparse Autoencoders: Interpretable Cross-Model Concept Alignment. Harrish Thasarathan, Julian Forsyth, Thomas Fel, Matthew Kowal, Konstantinos Derpanis
[3] Sparse Crosscoders for Cross-Layer Features and Model Diffing. Jack Lindsey*, Adly Templeton*, Jonathan Marcus*, Thomas Conerly*, Joshua Batson, Christopher Olah.

**Questions:**

- Why do the authors choose to max-pool over the sequences?
- The authors write the following: "Steering with single directions often produces more incoherent generations, like endless repetitions of similar tokens. We instead combine several semantically related features to obtain more robust concept directions." How are the several related features found, and how are they combined? If steering one single direction is ineffective, does that not mean that that concept itself is poor?
- How are multiple layers steered simultaneously? Why have the authors chosen to do this?

---

> ### Author Response · Authors · 2025-11-22
>
> We thank the reviewer for their time and detailed feedback. While we appreciate their view that the application of the Platonic Representation Hypothesis is interesting and that our steering results are promising, we respectfully disagree with the assessment of both the computational cost and the methodological novelty of our approach. We believe there are important misconceptions about the relative scale of SAE training versus our alignment method which, once clarified, underscore the contribution of Atlas-Alignment.
>
> **1. On Motivation and Computational Cost:** To the best of our knowledge, state-of-the-art SAEs on LLMs are trained on 500M to multiple billion tokens to achieve low reconstruction error and high sparsity (cf. SAEBench Karvonen et al. Appendix B Table 3. [1]; Scaling and evaluating sparse autoencoders, Gao et al. [2]) and require multiple GPU-days or weeks. After training, automated labeling requires collecting highly activating samples over large datasets of 10M–2B tokens (Paulo et al., Section 3.1 [3]; Puri et al. Appendix B [4]) and running an explainer model for every single SAE feature.
>
> Atlas-Alignment, by contrast, requires a single forward pass over ~15M tokens (500k samples) and a lightweight alignment step that can be run on CPU in minutes to explain any further latent space. The resulting linear mapping directly enables interpretability and steering for a new latent space without retraining or relabeling. **For any additional model, this reduces required processing of tokens by roughly one to two orders of magnitude.**
>
> Beyond compute, unsupervised SAE features vary across runs due to splitting and absorption [5], preventing stable cross-model correspondence. Atlas-Alignment establishes a fixed semantic reference: aligning any subject model to a single Atlas guarantees that the same Atlas index corresponds to the same concept across models. This removes the need to re-discover or re-interpret concepts for every model. As stated in the paper, the Atlas need not be SAE-based; any interpretable feature dictionary is compatible, though SAEs provide strongly monosemantic features.
>
> **2. Dataset Size and Low-Data Regimes:** We agree that data requirements matter. To assess this, we subsampled the alignment dataset at multiple sizes (1k–500k samples) and evaluated translation quality and semantic retrieval for Llama-3.1-8B-Instruct (layer 19) aligned to the Gemma Scope 16k Atlas. Covariance and Cross-Correlation degrade only mildly as data decreases. Orthogonal Procrustes remains the strongest method across nearly all scales (except at 1k), with improvements tapering off as more data is added. Even at 50k samples, Procrustes still outperforms all other methods. Our results show that alignment quality can be balanced against compute, and that substantially smaller datasets are often sufficient, further strengthening the argument for Atlas-Alignment’s comparatively low computational cost. The full results will be added to the paper by November 26th.
>
> **3. On Comparison to Prior Work:** [6] Lee et al. focus on geometric similarity and map specific known steering vectors between models. In contrast, Atlas Alignment enables full concept-dataset-free interpretability of the target model (semantic search etc.) in addition to targeted steering.
> [7, 8] Universal SAEs & Crosscoders: These are joint training methods. They require training new dictionary learning architectures from scratch (high cost). Atlas-Alignment is a post-hoc transfer method.
>
> **4. On Baselines and Native SAE Comparison:** The reviewer suggests comparing Atlas-based steering against "steering with SAEs trained for the target model." This comparison is conceptually difficult because SAEs are unsupervised. A "Deception" feature in the Atlas might not have a precise 1-to-1 equivalent in a natively trained SAE. To nonetheless address the intent of the critique, we add an experiment using supervised steering vectors. For selected concepts, we construct supervised steering vectors in both the subject model and the Concept Atlas. We then compare steering performance using (1) the subject model’s own supervised vector and (2) the Concept Atlas vector transferred through Atlas-Alignment. Due to current computational constraints, we cannot report results yet, but we will include the full experiment by November 29th.

---

> ### Author Response · Authors · 2025-11-22
>
> **Question about the Pooling Operation:** Pooling is needed because alignment operates on a single vector per sample and tokenization differs between the subject model and the Atlas. We use max-pooling as it captures the strongest concept feature activation in the Concept Atlas for every sequence. To evaluate alternatives, we compared max- vs. mean-pooling on Qwen-2.5 3B (layer 21). Mean-pooling substantially reduced translation quality for Procrustes and linear regression; Covariance and Cross-Correlation showed mixed effects; SemanticLens benefited slightly. Procrustes remained the strongest method under both pooling choices. Since results are not uniform across methods, we will extend the analysis to additional layers and models and include a more detailed discussion in the revised paper by November 28th.
>
> **Question about Multi-Concept and Multi-Layer Steering:** Related Atlas features can be found by embedding Atlas labels with a standard embedding model and retrieving semantically close features. We combine multiple features with all features weighted with 1. Full details on the features are in Appendix A.7.1.
> Multi-Layer Steering:  Because concepts appear differently across layers, we apply the same query to multiple layers by translating it into each layer’s latent space. Importantly, unsuccessful steering of a concept does not necessarily imply that the underlying feature is “poor.” A feature can be a strong detector of a concept in an intermediate MLP layer while still having limited causal influence on generation if that layer or concept is not strongly connected to the model’s output pathways.
>
> [1] SAEBench: A Comprehensive Benchmark for Sparse Autoencoders in Language Model Interpretability, Karvonen et al., ICML (2025)
> [2] Scaling and evaluating sparse autoencoders, Gao et al., OpenAI (2024)
> [3] Automatically Interpreting Millions of Features in Large Language Models, Paulo et al., ICML (2025)
> [4] FADE: Why Bad Descriptions Happen to Good Features, Puri et al., ACL Findings (2025)
> [5] Towards Monosemanticity: Decomposing Language Models With Dictionary Learning, Bricken et al., Anthropic (2023)
> [6] Shared Global and Local Geometry of Language Model Embeddings, Lee et al., COLM (2025)
> [7] Universal Sparse Autoencoders: Interpretable Cross-Model Concept Alignment, Thasarathan et al. (2025)
> [8] Sparse Crosscoders for Cross-Layer Features and Model Diffing, Lindsey et al. (2024)

---

> ### Author Response · Authors · 2025-12-01
>
> We have updated the manuscript to include:
>
> (i) comparisons of pooling methods (Section 4.3.1, Appendix A.5.2), and
> (ii) ablation study on dataset size (Section 4.3.1, Section 4.3.2, Appendix A.5.1, Appendix A.6.2).
>
> Unfortunately, we will not be able to complete the previously announced steering experiment within the rebuttal window. We will add these results and the analysis to the camera-ready version of the manuscript. To partially address the reviewer’s request in the meantime: Huang et al. [1] includes a closely related experiment that evaluates cross-model transfer of supervised steering vectors, albeit for a single alignment method. We encourage reviewers to refer to those results as an approximate point of reference.
>
> Thank you again for your suggestions.
>
> [1] Cross-model transferability among large language models on the platonic representations of concepts, Huang et al., ACL (2025)

---

### Official Review · Reviewer_bP4o · 2025-11-01

**Soundness:** 3
**Presentation:** 3
**Contribution:** 3
**Rating:** 8
**Confidence:** 3

**Summary:**

This paper proposes Atlas-Alignment, a method that constructs a human-understandable set of concepts (called Concept Atlas) from a foundation model and then aligns the representations from other models to the Concept Atlas for interpretation or steering. The Concept Atlas is created using sparse autoencoders. Then, a translation matrix between any model's representation and the Concept Atlas can be learned using any representational alignment method like Orthogonal Procrustes Translation. With this, they show that neurons representing similar concepts can be found across models and the models can be steered along those directions as well.

**Strengths:**

* The paper is fairly well-written and easy to follow.

* The proposed method is fairly simple, intuitive, seems novel and is effective.

* Qualitative and quantitative results seem to be fairly extensive (across models and datasets) and showcase the effectiveness of the method.

**Weaknesses:**

* The high-level technical novelty may be somewhat limited. For example, [W1] also proposes alignment of the representation space between two models, albeit for zero-shot classification. Or [W2] proposes linear alignment in a similar fashion but to produce individual neuron explanations for vision models. I do agree that Atlas Alignment's contributions are useful and noteworthy, but it would be good to discuss similarities and differences with these works given the high-level similarities.

* In the proposed approach, the Atlas representations are translated into the subject model representations. Why not the other way? Apply the translation to the subject representations. That would make more intuitive sense because the Atlas representations are the human-understandable ones and it would be more useful to convert the subject model representations into Atlas representations.

### References

[W1] Moayeri et al., "Text-To-Concept (and Back) via Cross-Model Alignment", ICML 2023

[W2] Oikarinen and Weng, "Linear Explanations for Individual Neurons", ICML 2024

**Questions:**

Please address the questions/comments in the weaknesses section.

---

> ### Author Response · Authors · 2025-11-22
>
> Thank you for engaging closely with the work and highlighting our extensive set of supporting evidence.
>
> **1. Relation to Prior Work:** We agree that these works share a high-level approach of learning mappings into a latent space. However, they differ significantly in their objectives and units of analysis. **Regarding [W1] (Text-to-Concept):** Their goal is **capability transfer**. They demonstrate that projecting an image encoder’s embeddings into CLIP’s space allows the model to inherit CLIP’s zero-shot classification abilities. In contrast, Atlas Alignment aims for interpretability and control. We do not seek to replace the subject model's capabilities, but to map its internal state to a legible atlas to facilitate understanding and editing.
> **Regarding [W2] (Linear Explanations):** This work is closely related but differs in the fundamental unit of interpretation. [W2] treats the individual neuron as the fixed atomic unit. Because neurons are often polysemantic, [W2] models them as sparse combinations (mixtures) of concepts. Conversely, Atlas Alignment treats the latent space as the unit, which lets us avoid training a separate sparse linear model for every neuron, as W2 does with GLM-Saga and greedy search. Instead of describing a fixed neuron as a mix of concepts, we can additionally search for directions in the latent space (linear combinations of neurons) that correspond to single, monosemantic concepts. While [W2] prioritizes descriptive fidelity of specific neurons, we prioritize identifying disentangled concept directions that also enable steering.
>
> **2. Clarification of Mapping Direction:** In our framework, the translation matrix can be used in either direction; the only asymmetry comes from the chosen alignment method. Linear regression and Orthogonal Procrustes produce mappings that reconstruct subject model activations from combinations of Atlas features, which matches the interpretation you suggest. Covariance- and correlation-based methods are symmetric and do not impose a direction. In all cases, the goal is to express each subject model feature in terms of the interpretable Atlas concepts. We will clarify this in the methods section.

---

### Official Review · Reviewer_A3qf · 2025-11-01

**Soundness:** 2
**Presentation:** 2
**Contribution:** 2
**Rating:** 4
**Confidence:** 3

**Summary:**

This paper introduces Atlas-Alignment, a framework that enables the transfer of interpretability between language models by aligning their latent spaces with a labeled Concept Atlas derived from sparse autoencoders (SAEs). Using lightweight representational alignment methods—particularly Orthogonal Procrustes—the framework allows an uninterpreted subject model to inherit semantic interpretability and controllable feature steering from an already interpreted model.

**Strengths:**

* The idea of transferable interpretability and connecting mechanistic interpretability with cross-model alignment theory is interesting.
* The paper provides comprehensive and strong experimental results.

**Weaknesses:**

* This work relies heavily on the "Platonic Representation Hypothesis". Would this simple linear alignment still hold if the "subject model" and "foundation model" differed significantly in architecture, scale, or training data distribution?
* The paper invokes the Platonic and Linear Representation Hypotheses as justification but does not provide a rigorous theoretical analysis of their validity.
* Max-pooling removes sequence structure, potentially weakening the interpretability of attention-related phenomena.
* Ablation study is missing so it's unclear how translation quality scales with the number.

**Questions:**

Please see the weaknesses.

---

> ### Author Response · Authors · 2025-11-22
>
> Thank you for the thoughtful review and for appreciating the connection of mechanistic interpretability and cross-model alignment in our work. We address your points below:
>
> **1. Alignment across Model Differences:** This is an excellent question. The extent to which linear alignment holds for very different models is fundamentally empirical, and our framework allows this to be tested directly rather than assumed. To address your point, we extended our translation-quality evaluation to additional subject models that vary substantially in scale, architecture, and tokenizer. We expanded the set to five additional models from the "Ministral", "Qwen 2.5", and "GPT-2" families to obtain a broader view of representational similarity across sizes, architectures, and tokenizers. The chosen models span a wide range: GPT-2 with 124M parameters; Qwen 2.5 with 0.5B, 1.5B, and 3B parameters; and the Ministral-8B-Instruct-2410 model with 8B parameters. We find that inter-model alignment scores (AUROC and AP) fall within the same range as the intra-model variation observed across layers of the Llama-3.1-8B models. Alignment is weakest for the smallest model (GPT-2 layer 8 MLP), similar to Llama 3.1 8B R1 Distill layer 12 MLP, and strongest for the 8B Ministral model, comparable to Llama-Base and Llama-IT layer 19. The relative strength of different alignment methods also transfers to these new models, indicating that alignment to the Concept Atlas arises largely from underlying representational similarity rather than dependence on the specific translation method. We will add the detailed experiment results to the paper by November 26th.
>
> **2. Theoretical Validity of the Platonic and Linear Representation Hypotheses:** These hypotheses serve as motivating intuitions for our framework; the method does not rely on them being universally true. Our contribution is empirical: the translation and retrieval evaluations directly test whether a linear mapping captures semantically meaningful structure for a given model pair. We will clarify this distinction in the revision and make explicit that the framework measures the extent to which these hypotheses hold rather than assuming them. Regarding the request for a rigorous theoretical analysis of the hypotheses: such an analysis is an open and active research direction in its own right and well beyond the scope of our paper. A formal introduction and treatment of these hypotheses is given in [1] and [2], and we follow that line of work rather than rederiving it here.
>
> **3. Pooling Operation:** Pooling is required because the alignment methods operate on a single activation vector per sample and because the subject model and the Concept Atlas may tokenize sequences differently. We adopt max-pooling since it reliably captures the strongest activation of a Concept Atlas feature independently of sequence length. To assess alternatives, we compared max- and mean-pooling for the Qwen-2.5 3B model. Mean-pooling yielded substantially lower translation performance (AUROC and AP) for Orthogonal Procrustes and linear regression. For the Covariance and Cross-Correlation methods, AUROC decreased while AP increased, and SemanticLens appeared to benefit from mean-pooling. However, Orthogonal Procrustes remained consistently stronger than all other translation methods under both pooling choices. Given the lack of a uniformly consistent trend, we will extend the analysis to additional models and layers and provide a more comprehensive discussion of pooling choices in the methods section. We thank the reviewer for raising this point and will add the detailed results to the paper by November 28th.
>
> **4. Impact of Dataset Size on Translation Quality:** To address this, we conducted a data-ablation experiment by subsampling the alignment dataset at multiple sizes (1k–500k samples) and evaluated the translation quality and semantic retrieval for Llama-3.1-8B-Instruct (layer 19) aligned to the Gemma Scope 16k Atlas. Covariance and Cross-Correlation degrade only mildly as data decreases. Orthogonal Procrustes remains the strongest method across nearly all scales (except at 1k), with improvements tapering off as more data is added. Even at 50k samples, Procrustes still outperforms all other methods. Our results show that alignment quality can be balanced against compute, and that substantially smaller datasets are often sufficient. Full results will be added to the paper by November 26th.
>
> [1] The platonic representation hypothesis, Huh et al., ICML (2024)
> [2] The linear representation hypothesis and the geometry of large language models, Park et al., ICML (2024)

---

> ### Author Response · Authors · 2025-11-27
>
> We have now updated the manuscript to include:
>
> (i) results for the additional subject models (Section 4.3.1, Appendix A.5), and
> (ii) comparisons of pooling methods (Section 4.3.1, Appendix A.5.2), and
> (iii) ablation study on dataset size (Section 4.3.1, Section 4.3.2, Appendix A.5.1, Appendix A.6.2).
>
> Thank you again for your suggestions. We are happy to provide further clarification if needed.

---

### Official Review · Reviewer_6tH6 · 2025-11-01

**Soundness:** 3
**Presentation:** 3
**Contribution:** 3
**Rating:** 6
**Confidence:** 3

**Summary:**

The paper proposes a method to map SAEs trained on one model to another "subject" model. To this end, they assume that reprenentations of different models converge to a single representation space, as has been discussed in previous work. They create an SAE concept atlas (the human-labeled activations) and align the hidden representations of the subject models to that atlas using training-free representational alignment methods.  They show that with this alignment matrix, the subject model inherits semantic feature search from the first model (queries for “chess,” “dreams,” or “reddit comments”), and also concept steering (injecting translated concept directions into its activations), becomes possible.
To this end, the paper evaluates five concept atlas translation methods covariance, correlation, linear regression, SemanticLens-style, and orthogonal Procrustes) from Gemma-2-2B-based Gemma Scope 16k and Gemma Scope 65k SAEs across several subject models (Llama 3.1 8B Base, Instruct, R1-Distill). It evaluates translation quality (AUROC/AP), semantic retrieval (MRR/MPP, near-perfect), and model steering, where Procrustes performs best, e.g., 30–40% faithfulness improvements over baseline model steering).

**Strengths:**

- Reusing SAEs and various architectures is highly interesting. First to enable cost-efficient reuse and also for model steering
- The paper evaluates various alignment methods and several tasks/applications

**Weaknesses:**

- The paper relies on only a single atlas from Gemma scope.
- It pools activations and as such loses some more specific information.
- A human evaluation would strengthen the approach
- Much of the method (e.g., cross-model alignment, universal SAEs) relies on previous work, see related work

**Questions:**

Have you tried aligning to a model with different architecture, tokenizer, model sizes?
How stable is steering when we use token-based atlases or using atlases from different models?

---

> ### Author Response · Authors · 2025-11-22
>
> Thank you for the thoughtful review and for highlighting the value of reusing SAEs across architectures and tasks. We address your points below:
>
> **1. Use of a single Concept Atlas:** You are correct that the experiments rely on the Gemma Scope Atlases. The availability of a fully labeled and thoroughly validated SAE is the main practical bottleneck for expanding this set. To nonetheless investigate how our method generalizes across different latent spaces, and to answer your question about using different subject models, we have extended the evaluation. Concretely, we have expanded the set of subject models to include five more models from three model families "Ministral", "Qwen 2.5" and "GPT-2", to provide a view of the representational similarity across sizes, architectures and tokenizers. The chosen models vary strongly in size, with GPT-2 containing 124M parameters, the Qwen 2.5 models with sizes of 0.5B, 1.5B and 3B respectively and Ministral-8B-instruct-2410 containing 8B parameters. Their alignment scores (AUROC/AP) lie within the same range as the intra-model variation observed across Llama-3.1-8B layers. GPT-2 shows the weakest alignment, similar to lower-performing Llama R1-Distill layers, while Ministral-8B shows the strongest, comparable to the best Llama-Base/IT layers. The relative ranking of alignment methods remains consistent across all models, suggesting that alignment performance is primarily driven by representational similarity rather than the specific translation technique. Full results will be included in the paper by November 26th.
>
> **2. Pooling operation:** Pooling is required because alignment methods operate on a single activation vector per sample, and models may tokenize sequences differently. We adopt max-pooling since it reliably captures the strongest activation of a Concept Atlas feature independently of a sequence's length. Nevertheless, assessing alternative pooling strategies is important. Accordingly, we conducted a comparison of max- and mean-pooling for layer 21 of Qwen-2.5 3B. Mean-pooling substantially reduced translation quality for Procrustes and linear regression; Covariance and Cross-Correlation showed mixed effects; SemanticLens improved slightly. Procrustes remained the strongest method under both pooling choices. Given the lack of a uniformly consistent trend, we will extend our analysis to additional models and layers and provide a more comprehensive discussion of pooling choices, which we will add to the paper by November 28th.
>
> **3. Need for Human Evaluation:** We agree with the reviewer that a human user study represents a valuable extension of our work. However, our current results already provide strong evidence for the validity and usefulness of Atlas-Alignment. The quantitative evaluations demonstrate high translation scores, accurate feature-retrieval metrics, and consistent steering performance, showing that the aligned features preserve the statistical structure of the Concept Atlas and that both semantic retrieval and concept-conditioned control behave as expected. The qualitative examples reinforce this: the semantic feature identification visualizations and the targeted steering outputs clearly show that aligned features activate on the intended concepts and that generations shift in predictable, concept-specific ways. Together, these results serve as a practical proxy for human utility. Given this combined evidence, we believe the paper already supports its core claims without requiring a human study in the current review cycle. We agree that a human evaluation would be a valuable next step. Ultimately, interpretability and control methods must prove their usefulness to human practitioners, and a targeted user study is therefore a high-priority item for future work.
>
> **4. Dependence on Cross-Model Steering and Universal SAEs:** Our main contribution is the transferral of interpretability from one interpretable reference space, the Concept Atlas, to another unknown latent space using a range of representational Alignment methods. This enables feature identification and dataset-free concept steering in previously unknown latent spaces. Cross-Model Steering [1] conversely investigates transferring supervised steering vectors across models using linear regression. We do not rely on Universal SAEs [2] in our work - any labeled SAE or sufficiently interpretable and monosemantic latent space can be used as Concept Atlas.
>
> **Question on “token-based” Atlas Steering:** We are unsure about the term "token-based atlases" within the context of our work, but if the reviewer is referring to highly monosemantic and localized features within the Concept Atlas, we have qualitatively found these features to perform well.
>
> [1] Cross-model transferability among large language models on the platonic representations of concepts, Huang et al., ACL (2025)
> [2] Universal Sparse Autoencoders: Interpretable Cross-Model Concept Alignment, Thasarathan et al. (2025)

---

> ### Author Response · Authors · 2025-11-27
>
> We have now updated the manuscript to include:
>
> (i) results for the additional subject models (Section 4.3.1, Appendix A.5), and
> (ii) comparisons of pooling methods (Section 4.3.1, Appendix A.5.2).
>
> Thank you again for your suggestions. We are happy to provide further clarification if needed.

---

### Official Review · Reviewer_3aNn · 2025-11-03

**Soundness:** 3
**Presentation:** 3
**Contribution:** 1
**Rating:** 4
**Confidence:** 3

**Summary:**

This paper introduces a method for aligning language models representations to discovered concepts in a separate language models.
By evaluating the linear transformation among two representation spaces, the resulting map transfers concepts of the SAE-based reference to the language models under study.

The paper is mainly experimental and investigates the efficacy in transferring the concepts, the effect of steering, and semantic retrieval only by using the reference concepts of the SAE.

**Strengths:**

The idea behind the method is quite simple and straightforward: assuming the Platonic Representation Hypothesis, the authors propose to find a map $T_{s \to c}$ that maps model representations to activated concepts in the SAE reference.
The method is quite simple and works good with orthogonal transformations.

Overall, the results show the efficacy of the method for the task the authors have identified, including retrieval and steering. Moreover, the paper is well-written and easy to follow.

**Weaknesses:**

As far as I know, studying representational alignment through linear invertible transformations is a well explored field, where the most direct comparison is with [1].
Because of this, I cannot see much novelty in the proposal, if just considering using it with language models. While the results show efficacy in transferring the concepts of the SAE to another language models, it is not clear how this differs from previous proposal in [1].

Moreover, the paper rests on two fundamental hypotheses which have not yet been verified: the Platonic Representation Hypothesis, that even in its full acceptance does not necessarily imply linear transformations among representations, and the Linear Representation Hypothesis. For the latter, I think it is not necessary to assume it to have the method to work, since transferring concept of the SAE can be done in a linear manner even if the SAE concepts are not interpretable.

Having accounted the gap in representational similarity between SAE reference and the subject models could be an interesting addition, which will make more concrete what limitations are inherited by the atlas. For similarity measures, I refer to [].

-----

[1] Latent Space Translation via Semantic Alignment, Maiorca et al. NeurIPS (2023) \
[2] Similarity of Neural Network Models: A Survey of Functional and Representational Measures, Klabunde et al., ACM (2023)

**Questions:**

I could not understand all details of the semantic retrieval setup that you tested.
In particular, why are "subject model features i" used for the test? Do you mean choosing individual neurons i of the language models and then select inputs that maximally activate them? How costly is this selection and why not using sentences with ground-truth concepts annotations (e.g., the verb tense or the truthfulness)? What is the ground-truth in this setting?

---

> ### Author Response · Authors · 2025-11-22
>
> We thank you for your positive feedback and detailed comments. We appreciate that you found the paper clear and the experimental results for retrieval and steering strong.
>
> **1. Relation to Prior Work:** The reviewer is correct that the use of lightweight, linear transformations for aligning neural representations is a known and established technique, as exemplified in [1]. However, the novelty of the Atlas-Alignment framework does not lie in the discovery of the underlying algebraic tool, but in its application and the specialized nature of the reference space we align to. Our core contribution is enabling the transfer of interpretability and control across LLMs, which requires fundamentally different considerations and validation than the component stitching proposed in [1]. More specifically, the works diverge significantly in their goals and methodology.
> Core Objective: Our primary goal is the transfer of mechanistic interpretability: enabling semantic search and controllable steering in new, opaque models without retraining and labeling of interpretability tools like SAEs. In contrast, [1] focuses on zero-shot component stitching, showing that transformations can reuse pre-trained encoders and decoders across models, domains, and modalities to improve performance on downstream tasks such as classification or reconstruction.
> Reference Space: We align a subject model’s latent space to a Concept Atlas, a structured, sparse, semantically labeled latent space derived from a SAE that serves as a dictionary of human-understandable concepts. In [1], space X is instead aligned to a generic latent space Y to allow Y’s classifier or decoder to operate on translated features. Core Novelty: Our main novelty is demonstrating alignment to this specific human-interpretable latent space, allowing the subject model to inherit labeled semantic meaning for analysis and steering. Whereas [1] uses linear alignment to transfer function, Atlas-Alignment uses it to transfer meaning and control. We will clarify this distinction in the revised introduction and related work.
>
> **2. Platonic and Linear Representation Hypotheses:** The reviewer raised concerns that our method depends on the platonic and linear representation hypotheses, of which a rigorous theoretical verification is still an area of active research. While there has been empirical evidence in recent literature that supports both hypotheses, this is a valid concern. However, in practice, the hypotheses serve mainly as a motivation for our framework and the framework does not rely on them being universally true. Instead, we can empirically measure how well concepts transfer between specific model pairs via the translation quality metrics. This allows us to evaluate the quality of alignment directly rather than assume it from the hypotheses.
>
> **3. Representational Similarity of Latent Spaces:** To address the reviewer’s point on representational similarity, we expanded our subject models to include three additional families: Ministral, Qwen 2.5, and GPT-2, and evaluated the alignment to the Gemma Scope Concept Atlas across sizes, architectures, and tokenizers. The models span a wide parameter range (GPT-2 at 124M; Qwen 2.5 at 0.5B, 1.5B, and 3B; and Ministral-8B-Instruct-2410 at 8B). We find that inter-model alignment scores (AUROC and AP) fall within the same range as intra-model variation across Llama-family layers. Alignment is weakest for the smallest model (GPT-2 layer 8 MLP), similar to Llama 3.1 8B R1 Distill layer 12, and strongest for the 8B Ministral model, comparable to the best Llama-Base and Llama-IT alignments at layer 19. The relative performance of different alignment methods also transfers to these new models. Overall, the results suggest that successful alignment to the Concept Atlas is driven primarily by underlying representational similarity rather than the specific translation method. We will add full results to the paper by November 26th.
>
> **Question on semantic retrieval experiment:** Regarding the question about the semantic retrieval experiment: Exactly, we choose at random a set of subject model neurons. We run an automated interpretability pipeline, as e.g. outlined in [2, 3] to create natural language descriptions, describing the concepts the neurons encode and validate the adequacy using an automated evaluation framework. This yields a set of ground-truth concept annotations for each of the chosen subject model neurons. We then create a set of samples for each of the ground-truth concepts, build a targeted concept-query, as outlined in sections 3.4 and 4.3.2, and measure how well we can recover the correct neuron from the subject model latent space using Atlas-Alignment.
>
> [1] Latent Space Translation via Semantic Alignment, Maiorca et al., NeurIPS (2023).
> [2]  Language models can explain neurons in language model, Bills et al., OpenAI (2023).
> [3] FADE: Why Bad Descriptions Happen to Good Features, Puri et al., ACL Findings (2025).

---

> > ### Comment · Reviewer_3aNn · 2025-11-24
> > **Reply**
> >
> > Thank you for the clarification and comparison with [1]. My point was indeed that using affine transformations to relate spaces is an already explored area which should be included in the intro and related work. Beyond model stitching, similar properties such as alignment can be obtained if affine transformations are found.
> >
> > The efficacy of both [1] and of Concept Atlas relies on the fact that finding these alignment transformations is possible, and I agree this does not heavily depend on Linear Repr. Hypothesis, but rather it must depend on similarity of representations, if the Platonic Repr. Hypothesis does not hold such an alignment should not be possible.
> >
> > Thank you for including additional baselines, I will check the manuscript when updated.
> >
> > **Q** I am curious about this automated procedures. To what degree can they give ground-truth annotations? Does it mainly depend on the language model used to annotate neurons activations?

---

> > > ### Author Response · Authors · 2025-11-26
> > >
> > > Thank you for the follow-up. We have now included the additional experiments.
> > >
> > > We agree that affine representational alignment is a well-explored area. Our contribution with Atlas-Alignment is demonstrating that aligning to a structured, interpretable reference space (the Concept Atlas) enables the transfer of interpretable semantics and direct steering of otherwise opaque LLMs. We will update the introduction and related work to make this more explicit and better reflect prior work in the final manuscript.
> > >
> > > Regarding the Platonic Representation Hypothesis: We agree that a practical precondition for our framework is a sufficient representational similarity between the specific subject model and Concept Atlas.
> > >
> > > Regarding your question on automated interpretability procedures: These pipelines assign natural language descriptions to features (e.g. neurons or SAE features) aimed at capturing their behavior or role in the network (e.g. reliably activating on certain tokens/topics or mediating specific downstream effects). The quality of the resulting descriptions depends on many factors (including, as you suggest, the explainer LLM), which is why evaluation frameworks are important. [1, 2] compare the quality of descriptions generated from different explainer LLMs and find that model capability correlates with description quality, although this trend appears to saturate. For further details, we kindly refer you to these works.
> > > In our semantic retrieval experiment, we use Transluce labels [3] and restrict our evaluation to features that pass FADE validation [2], ensuring we only include neurons with verified, high-quality descriptions. We use these labels because Atlas-Alignment can be used as a cost-efficient alternative to running a full automated interpretability pipeline per subject model, and we want to measure how well we can recover the kind of features those pipelines capture when they succeed.
> > >
> > > [1] Automatically Interpreting Millions of Features in Large Language Models, Paulo et al., ICML (2025).
> > > [2] FADE: Why Bad Descriptions Happen to Good Features, Puri et al., ACL Findings (2025).
> > > [3] Scaling Automatic Neuron Description, Choi et al., (2024).

---

### Meta-Review · Area_Chair_rWj8 · 2025-12-28

**Summary:**

The paper introduces Atlas-Alignment, a framework designed to transfer interpretability across language models by aligning latent spaces to a labeled Concept Atlas derived from SAEs. While reviewers appreciated the potential for cost-efficient interpretability and the extensive experimental results across various architectures, the consensus is that the technical novelty is limited. Multiple reviewers pointed out that linear representational alignment is a well-explored area, and the specific application to SAEs constitutes an incremental engineering step rather than a significant methodological advance. Furthermore, concerns persisted regarding the method's reliance on max-pooling, which risks discarding essential sequence information, and the validity of the underlying Platonic Representation Hypothesis when applied to diverse architectures. Although the authors provided additional ablation studies and pooling comparisons during the rebuttal, these efforts did not fully address the fundamental critiques regarding the method's differentiation from prior alignment works and the strength of the baselines. Therefore, I recommend rejection and encourage the authors to demonstrate stronger differentiation and submit to a future venue.

**Reviewer Concerns:**

Concerns addressed by the rebuttal:
- Generalizability across architectures and scales by Reviewer 6tH6 and Reviewer A3qf.
- Impact of dataset size by Reviewer A3qf and Reviewer 5Awd.
- Justification of pooling operations by Reviewer 6tH6, Reviewer A3qf, and Reviewer 5Awd.

Outstanding concerns:
- Limited technical novelty regarding the application of well-explored linear alignment methods  by Reviewer 3aNn.

**Reviewer Scores:**

It seems like that one reviewer is likely to raise their score from 4 to 6 and the other reviewers would maintain their score if they had been able to participate fully in the discussion.

---

### Decision · Program_Chairs · 2026-01-26

Reject